# The effects of plant-based dietary patterns on the risk of developing gestational diabetes mellitus: A systematic review and meta-analysis

Yu Zhu[1,2☯], QingXiang Zheng[2,3☯], Ling Huang[4], XiuMin Jiang [1,2]*, XiaoXia Gao[1], JiaNing Li[1], RuLin Liu[1]

1 The School of Nursing, Fujian Medical University, Fuzhou City, Fujian Province, China, 2 Fujian Maternity and Child Health Hospital College of Clinical Medicine for Obstetrics & Gynecology and Pediatrics, Fuzhou City, Fujian Province, China, 3 Fujian Obstetrics and Gynecology Hospital College of Clinical Medicine for Obstetrics & Gynecology and Pediatrics, Fujian Medical University, Fuzhou City, Fujian Province, China, 4 Fujian University of Traditional Chinese Medicine, Fuzhou City, Fujian Province, China

☯ These authors contributed equally to this work.
* jzc0427@163.com

**Data Availability Statement:** All relevant data are within the paper and its Supporting Information files.

## Abstract

### Background

The worldwide prevention of gestational diabetes mellitus (GDM) is a significant challenge. Plant-based dietary patterns are a series dietary habits that emphasized foods derived from plant sources more and from animal foods less. Now, no consensus exists on the effects of plant-based dietary patterns on the incident of GDM.

### Objective

This study aimed to estimate the effects of plant-based dietary patterns on the risk of developing GDM.

### Methods

This systematic review was conducted following the checklist of PRISMA. Six electronic databases including PubMed, Embase, Web of Science, China National Knowledge Infrastructure, Wangfang, and Chinese Scientific Journals Database were searched from inception to November 20, 2022. A fixed or random effect model was used to synthesize results of included studies. Then, subgroup analysis, meta-regression and sensitivity analysis were performed to assure the reliability and stability of the results.

### Results

Ten studies including 32,006 participants were identified. The results of this study showed that the better adherence to the plant-based dietary patterns was related to the lower risk of developing GDM (RR = 0.88[0.81 to 0.96], $I^2$ = 14.8%). The slightly stronger association between plant-based diets and the risk of developing GDM was found when healthy plant-

**Funding:** This work was supported by the Fujian Maternity and Child Health Hospital (grant numbers [YCXH 22-02]) and Joint Funds for the Innovation of Science and Technology, Fujian Province (grant numbers [2020Y9133]). However, the funder had no role in study design, data collection and analyses, decision to publish, or preparation of the manuscript.

**Competing interests:** The authors have declared that no competing interests exist.

based dietary pattern index was included in pooled estimate (RR = 0.86[0.79 to 0.94], $I^2$ = 8.3%), compared with that unhealthy one was included (RR = 0.90[0.82 to 0.98], $I^2$ = 8.3%).

## Conclusion

The plant-based dietary patterns are associated with a lower risk of developing GDM. Furthermore, healthy plant-based dietary patterns are more recommended than unhealthy one. It is significant to help medical staff to guide pregnant women to choose reasonable diets.

## Introduction

Gestational diabetes mellitus (GDM) refers to diabetes diagnosed during gestation that not detected glucose intolerance prior to pregnancy [1]. GDM affects about 5.8~12.9% of pregnancies globally depending on different diagnostic criteria [2,3] and leads several adverse obstetrical outcomes including hypertensive disorders, cesarean section, neonatal hypoglycemia, and macrosomia [4–6]. Besides above complications for GDM women, the incidence of type 2 diabetes mellitus of them is explicitly higher than normal women [7], with 7.5 times. For offspring delivered by GDM women, they are more remarkable to suffer reduced insulin sensitivity and pancreatic β-cell dysfunction after they are adult [8,9]. Given to the high prevalence and detriment of GDM, it is imperative to identify potentially modifiable risk factors of preventing GDM [10].

Diet plays an important role during gestation [11]. Maternal dietary status regulated nutrient availability for the placental and fetal growth, and ultimately affected the long-term health of the newborn [12]. Compared with single food or nutrient, dietary patterns are more likely to reflect individual dietary habits as we do not intake isolated nutrients [13]. Substantial evidences have demonstrated that there are strong associations between dietary patterns and developing GDM [14–17]. Western dietary pattern, characterized by a high intake of red meat, processed meat and fast food, has a significant association with a higher risk of developing GDM [18]. Conversely, the Mediterranean dietary pattern, characterized by a high intake of fruits, vegetables and whole-grain cereals, was strongly related to lower risk of developing GDM [19]. As a modifiable factor, dietary patterns have recently become a focus attention in preventing and curing GDM.

Plant-based dietary pattern is an umbrella term that refers to a habit that emphasized foods derived from plant sources more and from animal foods less, such as vegetarian diets [20,21]. Globally, the prevalence of adopting vegetarian diets varied according to different countries culture, whereas it was commonly less than 10% [22]. It indicated that current proportion of pregnant women who adhered to vegetarian diets may remain relatively small. The Academy of Nutrition and Dietetics advocated that vegetarian diets for pregnant women could be nutritionally adequate in pregnancy and result in positive maternal and infant outcomes [23], and this viewpoint had got partial support from several previous systematic reviews [24–27]. Practically, women's dietary patterns changed little from before to during pregnancy [28]. It is confirmed that plant-based dietary pattern is beneficial to decrease the risk of developing type 2 diabetes mellitus [29]. However, the conclusion that the effects of plant-based dietary patterns on developing GDM remains unclear. Therefore, in this systematic review and meta-analysis, we aimed to investigate the effects of plant-based dietary patterns on the risk of developing GDM.

## Materials and methods

### Registration of review protocol

This review was registered at PROSPERO International Prospective Register of Systematic Reviews (https://www.crd.york.ac.uk/PROSPERO/) with registration No. CRD42022374542 and was conducted according to the checklist of PRISMA [30].

### Eligibility criteria

The inclusion criteria were defined as following: (a) Population involving women with singleton pregnancy and without any acute or chronic diseases that impact dietary intake, such as cancer or kidney disease. (b) The exposure of interest was adherence to the plant-based dietary patterns, which generally were defined as diets that consuming higher consumption of planted-based foods and lower consumption or avoidance of animal-based foods. (c) The comparator depends on actual exposure. (d) The outcome of interest was the incident of GDM. (e) Only prospective cohort studies were included since recall bias in dietary ascertainment exists more likely in retrospective studies. The exclusion criteria were defined as following: (a) Several study types including any studies involving non-human subjects, experimental or quasi-experimental trials, cross-sectional studies, retrospective cohort studies, case-control studies, systematic reviews, case reports, and literature published as meeting abstracts. (b) Studies that multivariate-adjusted relative risks (RRs) or hazard ratios (HRs) with a 95% confidence interval were not reported.

### Assessment of plant-based dietary patterns

According to related studies that assessed dietary patterns [31,32], three methods mainly were used to define the plant-based dietary patterns: (a) prior designated diets that avoidance of animal-based foods, such as vegetarian diet and vegan diet; (b) a previous plant-based dietary indices scores; and (c) a posteriori factor analysis to extract dietary pattern.

Especially, vegetarian diets typically included plant-based foods including grains, legumes, nuts, seeds vegetables and fruits, and excluded all kinds of animal-based foods such as meat, meat products, poultry, seafood, mollusks and crustaceans [25]. Vegetarian diets usually contain two main directions: (1) lacto-ovo-vegetarian diets, which excluded meat but included dairy products, eggs and honey; (2) vegan diets, which excluded meat, dairy products, eggs and honey [25]. In this study, non-vegetarian diets or omnivorous diets were considered as the comparators of vegetarian diets/vegan diets.

When turns to previous plant-based dietary indices scores, Satija et al. [33] created a concept of plant-based dietary pattern indices to assess the adherence to plant-based dietary patterns including overall plant-based dietary indices (PDI), healthy plant-based dietary indices (hPDI), and unhealthy plant-based dietary indices (uPDI). Scoring was done according to quantile with the lowest food consumption receiving 1 point and quantile with highest food consumption receiving 5 points. For PDI, positive scores were assigned to plant food groups, and reverse scores were assigned to animal food groups. For hPDI, positive scores were assigned to healthy plant food groups (include whole grains, fruits, vegetables, nuts, legumes, vegetable oils and tea/coffee), while reverse scores were assigned to animal food groups and unhealthy plant food groups. For uPDI, positive scores were assigned to unhealthy plant food groups (included refined grains, potatoes and sweets/desserts), whereas reverse scores were assigned to animal food groups and healthy plant food groups. By above definition, the lowest PDI, hPDI and uPDI quantile, which indicated the poorest insistence on the plant-based dietary patterns, were used as a comparator of highest PDI, hPDI and uPDI quantile, respectively.

Also, for the included studies that used factor analysis to extract plant-based dietary patterns, the lowest quantile was used as a comparator.

## Information sources

Six electronic databases including PubMed, Embase, Web of Science, China National Knowledge Infrastructure (CNKI), Wangfang, and Chinese Scientific Journals Database (VIP) were searched using a strategy of Mesh-term combined with text-word. The following terms were conducted: 'plant-based', 'plant-based diet', 'vegetarian', 'vegan', 'vegetable', 'dietary fiber', 'dietary pattern', 'food pattern', 'gestational diabetes mellitus', 'gestational diabetes', and 'GDM'. All possible studies were retrieved within a range of published years from inception up to November 20, 2022, and specific search strategies for each database can be found in S1 Appendix. No language restriction was applied.

## Study selection

Two investigators (Zhu and Zheng) searched the relevant literature back to back. All literature identified with the search process were imported into EndNote X9 software, and duplicated studies were deleted automatically and manually. Two investigators also independently screened the titles and abstracts of potential studies that met the eligibility criteria, studies with uncertainties would be reviewed with a full-text screen. Any discrepancies arising during the selection process were resolved through discussion with the third researcher (Jiang).

## Data extraction

Two investigators (Zhu and Zheng) independently extracted the characteristics of studies that were preliminarily included with reference to an Excel collected sheet that was designed in advance. The following information of each study were collected: study ID defined by the first author and publication year, country, case of GDM and total number of participants, participants characteristics (including source, mean age, mean body mass index before or during pregnancy), method and time of dietary evaluation, dietary feature of being compared, GDM ascertainment, and adjusted confounders in the statistical models. We contracted authors when necessary data were unavailable in additional material sets. Final data extraction was formed based on a consensus between Zhu and Zheng.

## Assessment of risk of bias

Two investigators (Zhu and Zheng) independently assessed the quality of preliminarily included studies using the Newcastle-Ottawa Scale (NOS) (http://www.ohri.ca/programs/clinical_epidemiology/). NOS consists of eight items in three dimensions (selection, comparability, and outcome), with one point for each item. Particularly, for an item about confounders control, two points were got at most. Sum scores of items were rated high, moderate, and low quality, with scores of 8–9 points, 6–7 points, and less than 6 points, respectively. Any divergence regarding assessment was resolved by consulting with the third researcher (Jiang).

## Synthesis of result

The pooled results of multivariate-adjusted RRs or HRs from each study were analyzed using Stata MP 14.0 to estimate the effects of plant-based dietary patterns on the risk of developing GDM. Statistical heterogeneity was assessed using the $I^2$ statistic and visualized using the forest plots. For the $I^2$ statistic, $I^2 \leq 50\%$ and $I^2 > 50\%$ indicated low and high quality, respectively. If a substantial amount of statistical heterogeneity was detected, a random-effects model was

used. Then, subgroup analysis, meta-regression and sensitivity analysis were conducted to examine the source of high heterogeneity. Otherwise, a fixed-effects model was used. Subgroup analysis and meta-regression analysis were estimated for regulating effect sizes of the following possible variables: study country (the United States, China, India, Australia and Malaysia), method of dietary patterns assessment (prior designated diets, calculated plant-based dietary indices scores and factor analysis to extract dietary pattern) and period of dietary patterns investigation (before pregnancy and during pregnancy). Sensitivity analysis was conducted by removing one study from overall analysis each time. Moreover, for studies that reported the results before and after BMI adjustment, we included those data to assess the pooled results changes before and after BMI adjustment. In the studies that defined adherence to plant-based dietary patterns using dietary overall or healthy/unhealthy plant-based dietary indices, the relationship for overall plant-based dietary indices was estimated in the pooled result evaluation, the relationship for healthful/unhealthy plant-based dietary indices were estimated in a sensitivity analysis. Besides, sensitivity analysis was conducted through excluding any single study. Additionally, potential publication bias was estimated through Egger's and Begg's test, and was visualized using funnel plots if included studies were greater than 10.

In the studies that defined the plant-based dietary patterns by means of plant-based dietary indices, we conducted a dose-response meta-analysis by calculating restricted cubic splines with three fixed knots (set at 10, 50, and 90% for scores on the plant-based dietary indices) to estimate possible nonlinear association. Generalized least-squares regression with the Stata package "glst" was used to explore the log-linear dose-response slope and to combine individual studies. $P$ values<0.05 with two-side test were considered statistically significant.

## Results

### Study selection

Fig 1 illustrated the process of literature search and selection. A total of 1,447 articles from six databases were searched. Among that, 350 duplicated articles were removed automatically and manually. Next, 1,057 articles were excluded after examining the title and abstract, and then 30 articles were excluded after reviewing the full text. Finally, 14 records from 10 prospective cohort studies were included in the meta-analysis involving 32,006 participants [34–43]. Particularly, Yisahak et al. [35] reported two records that full vegetarian diet and semi-vegetarian diet from two investigations. They defined 'full vegetarian diet' as never ate meat, poultry and fish, or ate these foods less than once a month but had no restriction on fish; and 'semi-vegetarian diet' as ate meat, poultry and fish greater than once a month, but less than once a week. Chen et al. [36] reported three records that calculated an overall plant-based diet index, a healthful plant-based diet index, and an unhealthful plant-based diet index from three investigations. Besides, Yong et al. [37] showed two records plant-based diet in first and second trimester from two investigations.

Table 1 reported the characteristics of the included articles that most of which were cohort study with large samples. Totally, 32,006 women were included, who were with mean pre-pregnancy/pregnancy BMI ranging from 20.4 to 25.8 kg/m² and their age ranging from 26.7 to 32.3 years old. All studies were controlled confounders as possible, which was adjusted by confounding factors such as BMI, age, and parity. Meanwhile, five cohort studies were conducted in China, two cohort studies were performed in America, and else were implemented in other countries. Regarding the time for sticking to plant-based dietary, eight studies applied during pregnancy and two applied before pregnancy. For plant-based dietary patterns ascertainment, three approaches were found including predefined the vegetarian diets, calculated plant dietary indices, and performed factor analysis. Among the included studies, four records (Nurses'

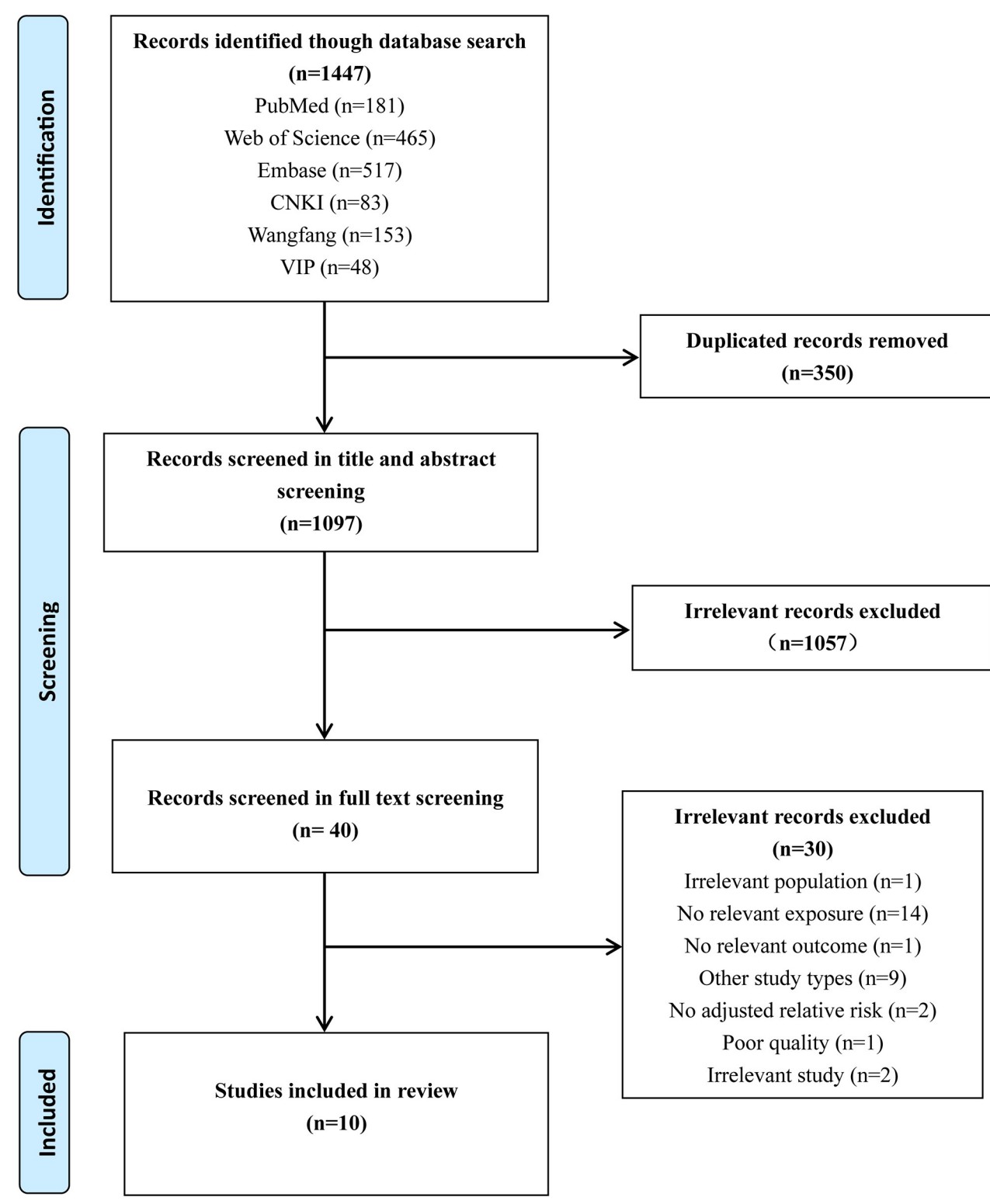

**Fig 1. PRISMA flow chart of study selection.**

**Table 1. The baseline characteristics of included studies.**

| Study ID | Country | GDM case/ total case | Age | Pre-pregnancy/ Pregnancy BMI | Source of population | Method of dietary patterns assessment | Period of dietary investigation | Method of GDM ascertainment | Exposure versus | Adjustment for confounders | NOS score |
|---|---|---|---|---|---|---|---|---|---|---|---|
| Schoenaker 2015 [34] | Australia | 292/ 3853 | 28.4±1.4 | 23.7~25.8 | Australian Longitudinal Study on Women's Health | FFQ, extracted dietary pattern using factor analysis | Before pregnancy | Self-reported that was demonstrated high agreement of 91% between self-reported GDM diagnosis in this study and administrative data records | "Cooked vegetable pattern", which was characterized by high consumption of carrots, peas, cooked potatoes, cauliflower, and pumpkin, comparing lowest tri quantile | Age, total energy intake, highest qualification completed, parity, parous, hypertensive disorders of pregnancy, polycystic ovary syndrome, inter-pregnancy interval, smoking, physical activity, and pre-pregnancy BMI. | 7 |
| Yisaha 2021 [35] | United States | NA/ 1859 | 27.7~32.1 | 23.4~25.4 | The *Eunice Kennedy Shriver* National Institute of Child Health and Human Development Fetal Growth Studies–Singletons | FFQ, predefined vegetarians and semi-vegetarians | During pregnancy | Defined using a combination of OGTT results based on Carpenter and Coustan criteria, which fasting plasma glucose>125 mg/dL or 2-h glucose >199 mg/dL , and medication treatment abstracted from medical records | Vegetarians, semi-vegetarians vs. non-vegetarians | Age, height, parity, pre-pregnancy BMI, race, marital status, education, income, current job/ student status, insurance coverage, infant sex, total weekly physical activity, total energy, and Healthy Eating Index-2010. | 8 |
| Chen 2021 [36] | United States | NA/ 14926 | 31.6~32.3 | 22.6~23.8 | Nurses' Health Study II | FFQ, calculated plant dietary index | Before pregnancy | Self-reported with confirmation by medical record review, diagnosis according to National Diabetes Data Group criteria | Overall plant-based dietary indices, healthy plant-based dietary indices, and unhealthy plant-based dietary indices, comparing lowest quintile | Age, parity, ethnicity, family history of diabetes, cigarette smoking, physical activity, alcohol intake, total energy intake, margarine intake, and pre-pregnancy BMI. | 7 |

*(Continued)*

**Table 1.** (Continued)

| Study ID | Country | GDM case/ total case | Age | Pre-pregnancy/ Pregnancy BMI | Source of population | Method of dietary patterns assessment | Period of dietary investigation | Method of GDM ascertainment | Exposure versus | Adjustment for confounders | NOS score |
|---|---|---|---|---|---|---|---|---|---|---|---|
| Yong 2020 [37] | Malaysia | 48/ 452 | 29.8~30.0 | 23.6~24.5 | The SECOST (Seremban Cohort Study) project | FFQ, extracted dietary pattern | During pregnancy, in first trimester and second trimester | According to Ministry of Health Malaysia guideline, diagnosis as GDM if fasting plasma glucose more than 5.6 mmol/l or 2h plasma glucose more than 7.8mmol/l in 75g OGTT | "Prudent, healthy pattern", which was predominantly plant-based with high factor loadings for other vegetables, nuts, seeds & legumes, green leafy vegetables, fruits, comparing lowest tri quantile | Clinic, gestational week at OGTT performed, maternal age, ethnicity, medical history of GDM, and family history of diabetes mellitus. | 8 |
| He 2015 [38] | China | 544/ 3063 | 28.9±3.2 | NA | Born in Guangzhou Cohort Study | FFQ, extracted dietary pattern | During pregnancy | Diagnosis GDM according to IADPSG criteria [a] | "Vegetable pattern", which was characterized by frequent intake of root vegetables, beans, mushrooms, melon vegetables, seaweed, other legumes, fruits, leafy and cruciferous vegetables, processed vegetables, nuts, and cooking oil, comparing lowest quartile | Other dietary patterns, age, education level, monthly income, pre-pregnancy BMI, family history of diabetes, and parity, parity. | 8 |
| Mak 2018 [39] | China | 199/ 1337 | 28.5±4.1 | 20.7±2.8 | Recruited from four maternity hospital | FFQ, extracted dietary pattern | During pregnancy | Diagnosis GDM according to IADPSG criteria [a] | "Plant-based pattern", which was characterized by high intakes of green leafy vegetables, cruciferous vegetables, gourd/melon family vegetables, red or orange vegetables, potatoes, root vegetables, bean vegetables, bean products, mushrooms, fruits, and low intake of lean pork meat, comparing lowest quartile | Age, family history of diabetes, parity, education level, and physical activity. | 8 |

*(Continued)*

**Table 1.** (Continued)

| Study ID | Country | GDM case/ total case | Age | Pre-pregnancy/ Pregnancy BMI | Source of population | Method of dietary patterns assessment | Period of dietary investigation | Method of GDM ascertainment | Exposure versus | Adjustment for confounders | NOS score |
|---|---|---|---|---|---|---|---|---|---|---|---|
| Zhou 2018 [40] | China | 248/ 2755 | 28.2±3.5 | 20.8±2.6 | Tongji Maternal and Child Health Cohort | FFQ, extracted dietary pattern | During pregnancy | Diagnosis GDM according to IADPSG criteria [a] | "Beans–vegetables pattern", which was characterized by including included higher intakes of root vegetables, melons and solanaceous vegetables, mushrooms and algae, beans, and bean products (soyabean, mung bean, soyabean milk), and leafy and cruciferous vegetables, comparing lowest quartile | Other dietary patterns, maternal age, ethnology, maternal education, average personal income, family history of diabetes, family history of obesity, smoking, alcohol, parity, pre-pregnancy BMI, weight gain before GDM diagnosis and total energy intake. | 9 |
| Wang 2021 [41] | China | 169/ 2099 | 28.0 (26.0~30.0) | 20.4 (18.8~22.2) | Tongji Maternal and Child Health Cohort | FFQ, calculated plant dietary index | During pregnancy | Diagnosis GDM according to IADPSG criteria [a] | Overall plant-based dietary indices, comparing lowest extreme quartile | Age, ethnicity, education, income, pre-pregnancy BMI, parity, family history of diabetes, smoking status, drinking status, exercise, sleep quality, weight gain before GDM diagnosis, intake of energy, juices, tea and coffee, sugar-sweetened beverages, and animal fat. | 9 |
| Mahendra 2022 [42] | India | 157/ 714 | 26.7±4.3 | 23.8 (20.4~26.5) | Recruited from antenatal clinics of two hospital | FFQ, extracted dietary pattern | During pregnancy | 139 cases were diagnosed as GDM according to 75g OGTT with confirmation by IADPSG criteria, and 18 cases were diagnosis as GDM according to glucose challenge test in which single blood glucose value (>140 mg/dl) 1h after a 50g glucose drink | "The healthy traditional vegetarian pattern", which characterized by devoid of any non-vegetarians and unhealthy food such as processed fried and fast food and sweet, followed by one unit of HTV dietary pattern score increased | Age, gravidity, polycystic ovarian syndrome status, area of residence, family type, family history of T2DM, use of nutrient supplements, physical activity, socio-economic status measures, and pre-pregnancy BMI. | 8 |

*(Continued)*

**Table 1.** (Continued)

| Study ID | Country | GDM case/ total case | Age | Pre-pregnancy/ Pregnancy BMI | Source of population | Method of dietary patterns assessment | Period of dietary investigation | Method of GDM ascertainment | Exposure versus | Adjustment for confounders | NOS score |
|---|---|---|---|---|---|---|---|---|---|---|---|
| Wang 2021 [43] | China | 991/ 1008 | 31±3.5 | NA | Recruited from antenatal clinics | FFQ, extracted dietary pattern | During pregnancy | Diagnosis GDM according to IADPSG criteria [a] | "Vegetable-fruit and potato pattern", which was characterized by high consumption of potatoes, fruits, vegetables, milk, and cereals | Other dietary patterns, age, ethnicity, education level, monthly income, family history of diabetes mellitus, smoking state, parity, pre-pregnancy BMI, gestational weight gain during OGTT, physical activity per week, energy intake. | 9 |

GDM, gestational diabetes mellitus; OGTT: Oral glucose tolerance test; IADPSG criteria, International Association of Diabetes and Pregnancy Study Group.

[a] IADPSG criteria, diagnosed women as GDM when any of the glucose values during the diagnostic OGTT: Fasting plasma glucose $\geq 5 \cdot 1$ mmol/l or 1-h plasma glucose $\geq 10 \cdot 0$ mmol/l or 2-h plasma glucose $\geq 8 \cdot 5$ mmol/l.

Health Study II a [36], Nurses' Health Study II b [36], Nurses' Health Study II c [36], and Tongji Maternal and Child Health Cohort [41]) reported plant-based dietary patterns using plant-based indices which developed by Satija[33], and three records [36] reported the effect size before and after BMI adjustment only. And two records (National Institute of Child Health and Human Development a [35] and National Institute of Child Health and Human Development b [35]) showed that individuals adherence to pre-defined vegetarian or semi-vegetarian diet compared with those who were not adherence to a vegetarian or semi-vegetarian diet. Plus, eight records [34,37–40,42,43] extracted dietary pattern performing factor analysis approach. Detailed scores of all cohort studies of moderate to high quality can be found in S2 Appendix.

## Plant-based diet and risk of developing GDM

We estimated the impact of BMI adjustment on the association between plant-based dietary patterns and the risk of developing GDM (Fig 2A). The pooled results indicated that the cumulative analysis for 14 records that greater persistence in plant-based dietary patterns was related to a lower risk of developing GDM, with fixed effects pooled effect size (RR = 0.88[0.81 to 0.96], $I^2$ = 14.8%). Fig 2B demonstrated that stronger correlation was found before adjusting BMI (RR = 0.85[0.78 to 0.92], $I^2$ = 28.3%). The result is still robust. We also assessed the impact of plant-based diet quality on the risk of developing GDM. The results indicated that overall RR slightly strengthened when "healthy plant-based dietary indices" were reserved in pooling RR (RR = 0.86[0.79 to 0.94], $I^2$ = 8.3%) (Fig 3A), however, overall RR weakened when "unhealthy plant-based dietary indices" were reserved (RR = 0.90[0.82 to 0.98], $I^2$ = 8.3%) (Fig 3B).

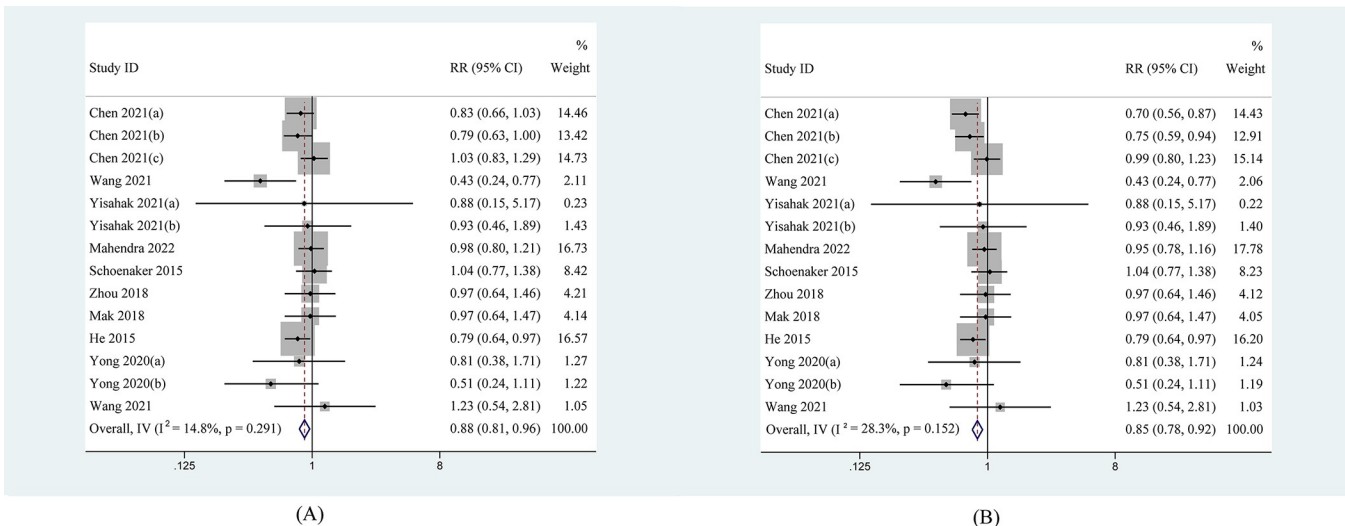

**Fig 2. Forest plot estimating the association between plant-based dietary patterns and risk of gestational diabetes mellitus.** (A) After BMI adjustment; (B) Before BMI adjustment.

## Dose-response relationships between plant-based diet and risk of developing GDM

To further assess the relationship between plant-based diets and the risk of developing GDM, we performed a quantitative exploration on four records from two studies [36,41] that expressed plant-based diets by plant dietary indices using fixed model. The higher plant dietary indices indicated greater adherence to plant-based diets. Fig 4 showed that the plant dietary indices are negatively associated with the risk of developing GDM, with a linear trend ($P$ = 0.018 for linearity and $P$ = 0.064 for nonlinearity).

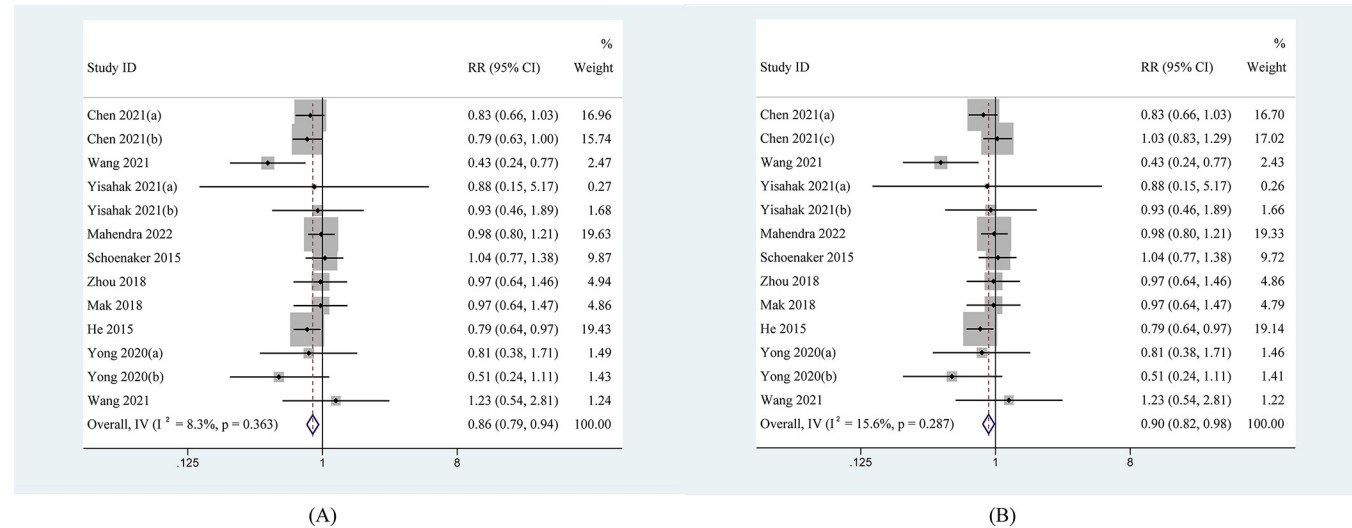

**Fig 3. Forest plot estimating the association between plant-based dietary patterns and risk of gestational diabetes mellitus.** (A) When healthy plant-based index was included; (B) When unhealthy plant-based index was included.

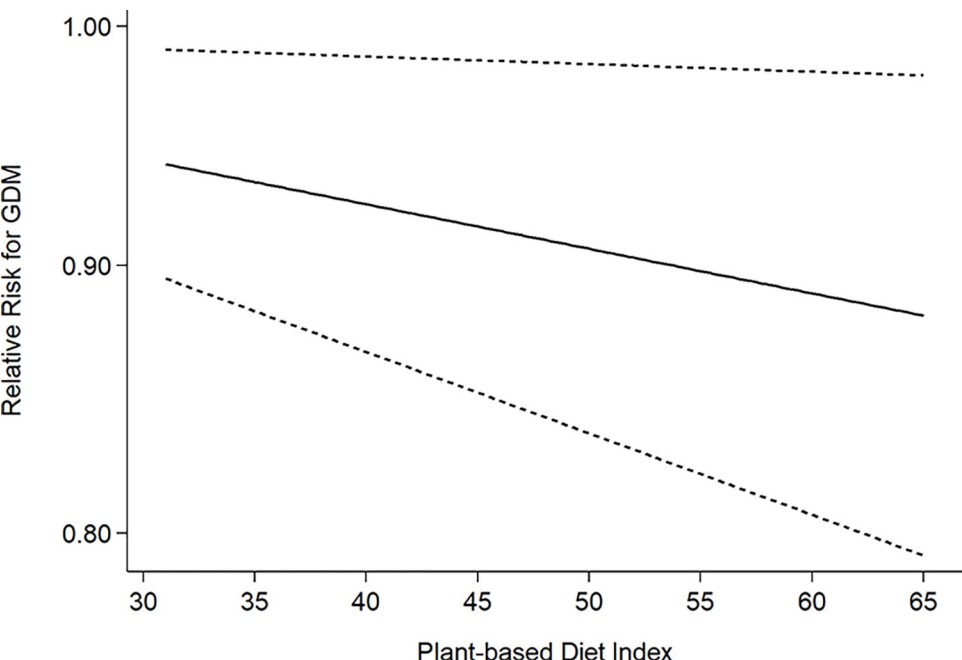

**Fig 4. Linear dose-response adherence to plant-based dietary patterns and risk of developing gestational diabetes mellitus.**

## Subgroup analysis of the effects of plant-based dietary patterns on the risk of developing GDM

No evidence in this study manifested remarkably high heterogeneity among included studies. To evaluate exposure stratification of interest, we performed subgroup analysis based on study country, method of dietary pattern assessment, and period of dietary investigation (Table 2). For study country, the results revealed that the conclusion remains invariant in studies conducted in China (RR = 0.82[0.70 to 0.96]) and the United States (RR = 0.88[0.78 to 1.00]), however, no significant association can be found between plant-based dietary patterns and the risk

**Table 2. Subgroup analysis of plant-based dietary patterns and gestational diabetes mellitus.**

| Subgroup | Numbers of Records | Pooled Relative Risk (95%CI) | $I^2$ (%) | P Value for Heterogeneity Between Subgroups |
|---|---|---|---|---|
| Main Estimate | 14 | 0.88 (0.81 to 0.96) | 14.8 | 0.29 |
| **Study Country** | | | | |
| the United States | 5 | 0.88 (0.78 to 1.00) | 0 | 0.54 |
| China | 5 | 0.82 (0.70 to 0.96) | 43.1 | 0.14 |
| India | 1 | 0.98 (0.80 to 1.21) | None | None |
| Australia | 1 | 1.04 (0.78 to 1.39) | None | None |
| Malaysia | 2 | 0.65 (0.38 to 1.10) | None | None |
| **Method of Dietary Patterns Assessment** | | | | |
| Calculated plant-based dietary indices | 4 | 0.85 (0.75 to 0.97) | 65.1 | 0.04 |
| Predefined vegetarians and semi-vegetarian | 2 | 0.92 (0.48 to 1.78) | 0 | 0.96 |
| Extracted dietary patterns via factor analysis | 8 | 0.91 (0.81 to 1.02) | 0 | 0.53 |
| **Period of Dietary Patterns Investigation** | | | | |
| Before Pregnancy | 4 | 0.91 (0.80 to 1.02) | 26.9 | 0.25 |
| During Pregnancy | 10 | 0.86 (0.76 to 0.97) | 16.9 | 0.29 |

of GDM from studies that conducted in other countries (India: RR = 0.98 [0.80 to 1.21], Australia: RR = 1.04 [0.78 to 1.39], and Malaysia: RR = 0.65 [0.38 to 1.10]). Regard to the method of dietary pattern assessment, association was detected only between the risk of developing GDM and plant-based dietary patterns that were defined by plant-based dietary indices (RR = 0.85 [0.75 to 0.97]). When turn to the period of dietary investigation, adherence to plant-based dietary patterns during pregnancy was passively related to the risk of developing GDM (RR = 0.86 [0.76 to 0.97]), but adherence to plant-based dietary patterns before pregnancy was not significantly associated with GDM (RR = 0.91 [0.80 to 1.02]).

## Meta-regression of the effects of plant-based dietary patterns on the risk of developing GDM

Meta-regression results illustrated that there was no relationship between plant-based dietary patterns in different country and the risk of developing GDM (coefficient estimate = 1.14 [-0.88 to 1.11], $P$ = 0.775). Meta-regression found no significant association between the risk of developing GDM and the plant-based dietary patterns ascertained by different methods (coefficient estimate = 0.42[-0.07 to 0.16], $P$ = 0.443). Meta-regression examined the impact of the period of plant-based dietary patterns adherent to on the risk of GDM which indicated no relationship (coefficient estimate = -0.60[-0.29 to 0.17], $P$ = 0.579).

## Sensitivity analysis of the effects of plant-based dietary patterns on the risk of developing GDM

Considering such different conclusions across included studies, sensitivity analysis was conducted through excluding any single study. The results showed that no significant influence of any study on the overall effect size of the meta-analysis. Visual inspection can be found in Fig 5.

## Publication bias

The Funnel plot in Fig 6 displayed that all included studies were arranged symmetrically around the centerline, signifying no publication bias. Egger's test ($P$ = 0.44) and Begg's test ($P$ = 0.50) demonstrated that no potential publication bias was detected.

## Discussion

### Main findings

In this systematic review and meta-analysis, we included ten studies with moderate to high quality involving fourteen records and found that greater adherence to plant-based dietary patterns is inversely associated with the risk of developing GDM. Such association is stronger when healthy plant-based indices were included rather than unhealthy indices. Meanwhile, such relationship is stronger before BMI adjustment.

### Interpretations

The correlation between the plant-based dietary patterns and the lower risk of developing GDM may be explained by several mechanisms. First, plant-based dietary patterns are characterized by the high consumption of vegetable, fruit, whole wheat, and soy, with abundant dietary fiber and polyphenols. Previous studies showed that those components were associated with a lower risk of GDM [44–47]. Meanwhile, plant-based dietary patterns emphasized low consumption of red meat and ultra-processed foods which increased the risk of GDM as well

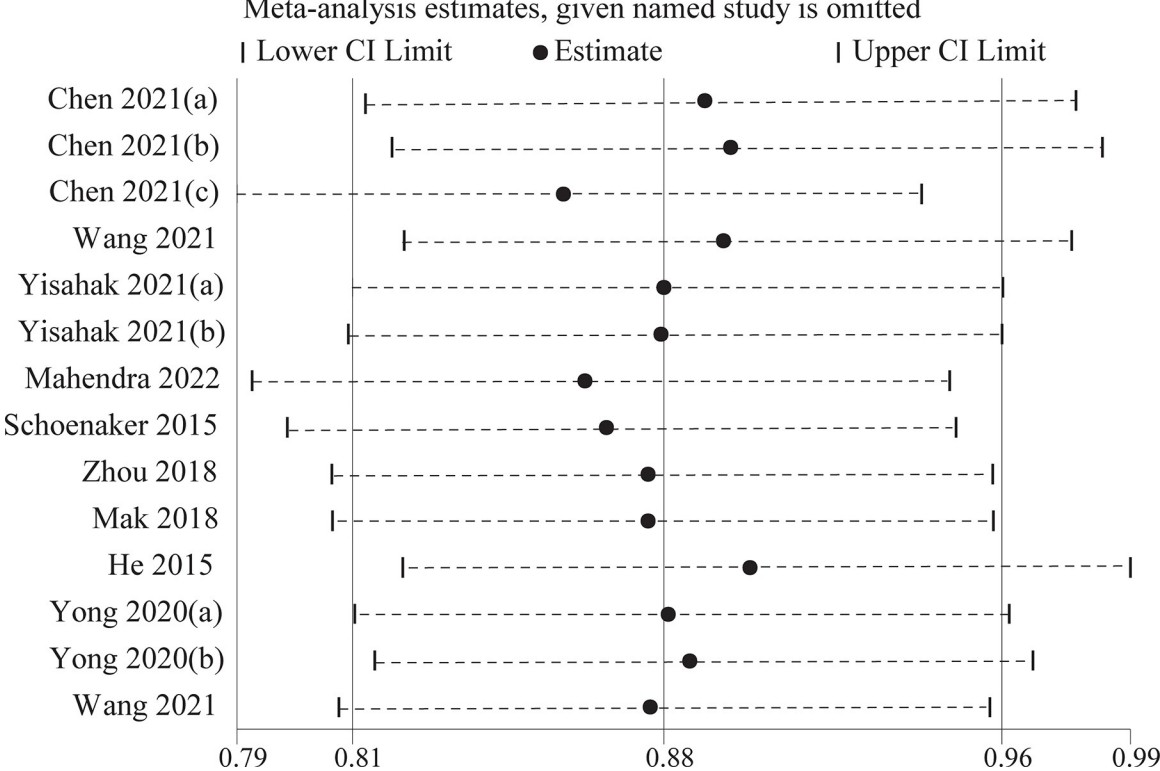

**Fig 5. Sensitivity analysis of the association between plant-based dietary patterns and risk of gestational diabetes mellitus.**

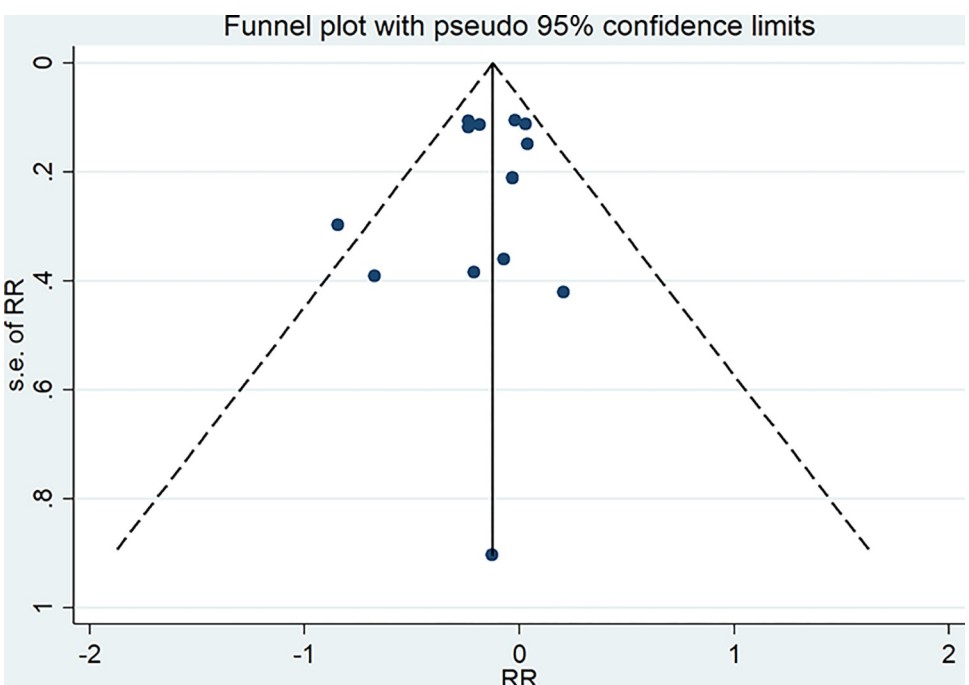

**Fig 6. Funnel plot of estimating the association between plant-based dietary patterns and risk of gestational diabetes.**

[18,48]. By evaluating the synergistic effects of nutrients through dietary pattern analysis, the impact of actual dietary intake on diseases was reflected. Second, plant-based diets may ameliorate GDM though alleviating insulin resistance and inflammation [49]. Insulin resistance and low-grade inflammation were recognized as the major etiologic characteristics of GDM. With moderate certainty of evidence demonstrated that a low-fat plant-based diet significantly improved insulin resistance and beta cell dysfunction [50–53]. Additionally, short-chain fatty acids (SCFA), fermentation products of dietary fiber, play a protective role in strengthening the colonic barrier, regulating intestinal function and reducing inflammation [54]. Among them, butyrate enhanced protection against inflammation and diabetes via extracellular regulated protein kinase (ERK) activation [55]. Third, the microbiome research provides a new idea. Previous studies illustrated that increase in the relative abundance of some microbiome during pregnancy are potentially related to the occurrence of GDM, such as *Collinsella*, *Rothia* and *Desulfovibrio* [21,56]. High dietary fiber intake can control overgrowth of *Collinsella* and promote SCFA-producing bacteria, in turn improving overall fermentation process in the gut [57]. Although no consensus has been reached on the causal link between diet, microbiota and disease, it is a promising strategy to prevent GDM by remodeling gut microbiota composition through diet modification [58,59]. Further investigation is needed to explore the complex interaction between dietary compounds and GDM in terms of microbiota and metabonomics.

To assess the consistency of the finding in different stratification, we performed subgroup analysis based on different study countries, method of dietary pattern assessment, and period of dietary investigation. For study countries, no significant relationship between GDM and plant-based diets in India, Australia, and Malaysian was revealed, which may be explained by that only a single study was included from these countries. For method of dietary pattern assessment, significant association between the plant-based diets and the lower risk of GDM only when ascertaining dietary pattern via plant dietary indices. It may be related to the subjective judgment of else two methods when extracting dietary patterns [60]. For the period of dietary investigation, the risk of GDM decreased when adhering to plant-based dietary patterns during pregnancy but not before pregnancy, which may be interpreted by that pregnant women whom adherence to plant-based diets before pregnancy do not necessarily continue it during pregnancy. Meta-regression found different stratification have no impact on the conclusion that the plant-based dietary patterns decrease the risk of developing GDM. In general, subgroup analysis and meta-regression indicated that the conclusion in this study is relatively stable and reliable.

The stronger pooled RR was observed when we considered "healthful plant-based dietary indices" (fixed-effects, RR = 0.86) instead of "unhealthy plant-based dietary indices" (fixed-effects, RR = 0.90) in an included study [36]. The finding indicated an opinion that not all plant-based diets were beneficial, which has been emphasized in prior studies [61,62]. Carbohydrates quality, a multidimensional index, usually be represented as some characteristics: glycaemic index, glycaemic load and dietary fiber content. It was considered as a crucial facet of plant-based diets [63]. For instance, plants with high glycaemic index, including potatoes, rice and refined bread, induced a sharp increase in glycaemia level that manifested the lower carbohydrates quality [64]. Plant-based diets dominated by such components were associated with higher risk of developing GDM [65,66], whereas plant-based diets with low glycaemic index were related to the lower prevalence of GDM [67]. Besides, Looman et al. [62] unexpectedly found that low-carbohydrate diets, characterized by high fat, high protein and low carbohydrates, were positive associated with the risk of developing GDM. Their further exploration revealed that women with higher fiber and fruits intake had a decreased risk of developing GDM, however, women with higher cereal intake had increased risk of developing GDM. It also underscored the importance of the carbohydrates source. Except for carbohydrates

quality, the source of protein and fat in plant-based dietary patterns mattered, too [68,69]. Pregestational low-carbohydrate diets with high protein and fat from animal foods were positively associated with the risk of developing GDM, nevertheless, low-carbohydrate diets with high protein and fat from vegetable foods were inversely related to the risk of developing GDM [70]. In detail, higher animal protein intake would increase the risk of developing GDM, whereas higher plant protein intake would do not [68]. In addition, higher fat intake, whether animal fat or plant fat, would increase the risk of developing GDM [69]. As such, plant-based dietary patterns with higher carbohydrates quality and plant protein consumption were more encouraged. Moreover, the concept of healthful plant-based diet was proposed by Satija et al. [33], who classified sugar-sweetened beverages, sweets and desserts, refined grains and juices as relatively unhealthy plant diets. These food groups were related to a higher risk of GDM [71], which may be interpreted by following potential mechanisms. Higher glycemic load from sugar-sweetened beverages and sweets and desserts may lead to increased weight gain through decreasing satiety or increased hunger signals, and disrupt metabolism via affecting neurochemical changes in the brain [72–74]. Milling and squeezing during the productive process of refined grains and juices reduced dietary fiber and bioactive phytochemicals, then potentially lowering dietary quality [75]. Significantly, it is necessary to distinguish whether the type of juice is homemade or commercial. The homemade juices might not be removed the pulp and edible skin, thus can provide some essential components for preventing GDM, such as fiber, vitamins, minerals, and phytochemicals [76]. Adversely, commercial juices provide only excessive sugars and energy that overload the glycometabolism. Overall, only healthy plant-based diets reduce the risk of GDM rather than unhealthy plant-based diets. We recommended that quality of foods should be emphasized when sticking to plant-based dietary patterns.

In this systematic review, we revealed that controlling weight may be a pathway through which plant-based diets reduce the risk of developing GDM [77,78]. Among included studies, three records from one study [36] reported the risk estimates before and after only BMI adjustment. The overall RR modestly strengthened when we included risk estimates before BMI adjustment, which demonstrated BMI acts as both a confounder and a mediator, which was consistent with previous findings [29]. All included studies adjusted BMI before or during pregnancy in statistical analyses, the underestimate of the effect of plant-based dietary patterns on the risk of developing GDM can be explained. Indeed, compared with BMI, gestational weight gain (GWG) is more important for pregnant women as energy for fetal growth is all from maternal nutrition during the whole pregnancy [12,79]. Regretfully, no GWG adjustment was particularly emphasized in our included studies, resulting in the exploration of the mediation of GWG can't be conducted. As such, we strongly suggest more evidences exploring the intermediary role of GWG.

## Strengths and limitations

To our best knowledge, this was the first meta-analysis to investigate the association between adherence to plant-based dietary patterns and the risk of developing GDM. Our review has several strengths. The majority of included studies were from large cohorts, which included numerous participants. High-quality studies that controlled as many confounders as possible were included. Additionally, we broadly synthesized plant-based dietary patterns defined by multiple analytic methods, and then further evaluated the robustness of the finding through subgroup analysis, meta-regression, sensitivity analysis, and dose-response analysis.

Several limitations in this study that merit consideration. First, measurement error is possible due to dietary patterns were estimated using a food frequency questionnaire by self-reported. Thus, only prospective cohort studies were included because the time point of

dietary investigation was closer to its actual implementation, to control further bias. Second, as previously demonstrated that not all studies adjusted for GWG in statistical analyses that we viewed it as an important mediator. We can't further explain the effect of GWG since deficiency of data and it needs to be specially observed in future assessments. Last, this review preliminarily validated the negative association between GDM risk and healthy plant-based diets instead of an unhealthy one. However, only one study discussed the quality of diet with low-intensity demonstration. The evidence that categorized by diet quality or explicit food groups are needed.

## Conclusions

Our systematic review and meta-analysis found that greater adherence to the plant-based dietary could reduce the incidence of GDM. Moreover, healthy plant-based dietary patterns may further decrease the risk of developing GDM than unhealthy plant-based dietary patterns. More explorations that focusing on mechanism of plant-base diets works and explicit food groups of healthy plant-based dietary patterns are required.

## Supporting information

**S1 Appendix. Search strategies for each database.**
(DOCX)

**S2 Appendix. Assessment of individual study bias.**
(DOCX)

**S3 Appendix. PRISMA checklist.**
(DOCX)

## Author Contributions

**Formal analysis:** Yu Zhu, QingXiang Zheng, Ling Huang.

**Methodology:** Yu Zhu, QingXiang Zheng, XiuMin Jiang.

**Software:** Yu Zhu, QingXiang Zheng, XiaoXia Gao.

**Validation:** XiaoXia Gao.

**Visualization:** Yu Zhu, QingXiang Zheng, XiuMin Jiang.

**Writing – original draft:** Yu Zhu, QingXiang Zheng, Ling Huang.

**Writing – review & editing:** Yu Zhu, QingXiang Zheng, Ling Huang, JiaNing Li, RuLin Liu.

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
