## [Decision Letter · Decision Letter 0]

5 Jul 2023

PONE-D-23-15568The effects of plant-based dietary patterns on the risk of developing gestational diabetes mellitus: A systematic review and meta-analysisPLOS ONE

Dear Dr. jiang,

Thank you for submitting your manuscript to PLOS ONE. After careful consideration, we feel that it has merit but does not fully meet PLOS ONE’s publication criteria as it currently stands. Therefore, we invite you to submit a revised version of the manuscript that addresses the points raised during the review process.

Please submit your revised manuscript by Aug 19 2023 11:59PM. If you will need more time than this to complete your revisions, please reply to this message or contact the journal office at plosone@plos.org. Please include the following items when submitting your revised manuscript:A rebuttal letter that responds to each point raised by the academic editor and reviewer(s). You should upload this letter as a separate file labeled 'Response to Reviewers'.A marked-up copy of your manuscript that highlights changes made to the original version. You should upload this as a separate file labeled 'Revised Manuscript with Track Changes'.An unmarked version of your revised paper without tracked changes. You should upload this as a separate file labeled 'Manuscript'.If applicable, we recommend that you deposit your laboratory protocols in protocols.io to enhance the reproducibility of your results. Protocols.io assigns your protocol its own identifier (DOI) so that it can be cited independently in the future. For instructions see: https://journals.plos.org/plosone/s/submission-guidelines#loc-laboratory-protocols. Additionally, PLOS ONE offers an option for publishing peer-reviewed Lab Protocol articles, which describe protocols hosted on protocols.io. Read more information on sharing protocols at https://plos.org/protocols?utm_medium=editorial-email&utm_source=authorletters&utm_campaign=protocols.

We look forward to receiving your revised manuscript.

Kind regards,

Kent Lai

Academic Editor

PLOS ONE

Journal Requirements:

"The authors declare that they have no competing interests."

Additional Editor Comments:

Some of the figures were quite blurry.  Please improve the resolution.

Reviewers' comments:

Reviewer's Responses to Questions

**Comments to the Author**

1. Is the manuscript technically sound, and do the data support the conclusions?

Reviewer #1: Yes

Reviewer #2: Yes

Reviewer #3: Partly

2. Has the statistical analysis been performed appropriately and rigorously? 

Reviewer #1: Yes

Reviewer #2: No

Reviewer #3: Yes

3. Have the authors made all data underlying the findings in their manuscript fully available?

Reviewer #1: Yes

Reviewer #2: Yes

Reviewer #3: Yes

4. Is the manuscript presented in an intelligible fashion and written in standard English?

Reviewer #1: Yes

Reviewer #2: Yes

Reviewer #3: Yes

5. Review Comments to the Author

Reviewer #1: Zhu and colleagues conducted a meta-analysis exploring the evidence linking plant-based or vegetarian dietary patterns to the risk of developing gestational diabetes mellitus. Ten prospective cohort studies were found in multiple databases and selected for the analysis. After robust statistical analysis which included assessment of the dietary patterns (plant based indices verses adherence to predefined vegetarian/vegan diets) and adjustments for confounders such as BMI, the authors concluded that plant-based dietary patterns are associated with a lower risk for developing gestational diabetes. This association was stronger when "healthy plant-based indices" were used, and weaker after adjusting for BMI. The authors address the confounder of BMI and gestational weight gain, the latter of which was not included in the analysis or the selected studies. The authors also speculate the potential mechanisms of which plant-based diets decrease the risk for gestational diabetes; one being the impact of changing the gut microbiome with increased consumption of plant-based foods and the decrease for inflammation, which is associated with diabetes.

Line 50-Led to should be "leads"

Line 52: Macrosomia and large for gestational age seem redundant; perhaps pick one to mention.

line 292: adversely might be changed to inversely

Overall, the authors did a thorough statistical analysis from various angles (heterogeneity, NOS scores, stratifying dietary indices etc) to assess the strength of the studies included. The results and interpretation are nicely described with good discussion of confounding factors.

Reviewer #2: In this paper, Yu Zhu and colleagues have done a systematic review and meta analysis on the effect of plant based dietary patterns on the risk of GDM. They have included 10 studies which included 32,006 participants. The results of the systematic meta analysis suggest that better adherence to plant based dietary patterns was related to lower risk of developing GDM. The authors have to be congratulated on a nice piece of work. This study has been done carefully. It has been registered at PROSPERO International Prospective Register of Systematic Reviews. Data extraction and assessment of risk of bias has been done carefully. The PRISMA flowchart of study selection presented clearly. Publication bias was tested by Egger's and Begg's test.

Suggestions :

It would have been nice if the composition of the plant based diets were examined in greater detail (for example, the breakdown of carbohydrate, protein and fat). Many plant based diets tend to be rich in carbohydrates which increase the risk of type 2 diabetes. In this context, if the authors could comment on why plant based diets are protective and how they differ from animal based diets, that would make the paper more interesting.

In the ‘Discussion” on Page 22 and in the “Conclusion”, they do talk about this. But perhaps in the context of healthy carbohydrates and unhealthy carbohydrates, this section can be expanded a little bit more.

Reviewer #3: I have several comments and suggestions regarding the methods section of the manuscript, specifically related to improving its clarity and comprehensibility. Please find below my detail comments:

1. To enhance clarity, several points in the abstract require further clarification:

a. Line 37-39: revisit the statement as the statement implies that both unhealthy and healthy plant-based diets are associated with a lower risk of GDM, with the healthy plant-based diet showing a slightly stronger association.

b. Line 40-41: This statement should be included in the methodology section.

2. Introduction:

a. While the author highlights the benefits of a plant-based dietary pattern and its positive effects on maternal and infant outcomes, it would be valuable to consider mentioning the current prevalence or proportion of individuals adopting such a dietary pattern. Including this information would help support the rationale for conducting this study and shed light on the relevance and potential impact of investigating the association between plant-based diets and gestational diabetes mellitus.

3. Materials and methods:

a. Line 92: The statement "Vegetarian dietary patterns or vegan dietary patterns were also reckoned with" may not conform to typical scientific writing standards.

b. Line 89-95: It would be beneficial to provide explicit definitions of both vegetarian and vegan dietary patterns to ensure clarity in understanding their distinctions. While both dietary patterns involve a focus on plant-based foods, vegetarian diets typically exclude meat, poultry, and seafood, while vegan diets abstain from all animal-derived products, including dairy, eggs, and honey. By clearly defining these terms, readers will have a better understanding of the specific dietary restrictions and considerations associated with each pattern, which is crucial for interpreting the study findings accurately.

c. Line 109: Please verify the accuracy of the term 'VIP' as a database name. Please double-check its validity or provide additional clarification if it refers to a specific database or acronym.

d. Please consider rechecking the PRISMA checklist to ensure its adherence to the guidelines. Upon review, it appears that certain items may not fully align with the recommended criteria. For example: Data items, list and define all outcomes for which data were sought. Specify whether all results that were compatible with each outcome domain in each study were sought (e.g. for all measures, time points, analyses)

e. Line 148-152: please provide additional elaboration on the details of the subgroup analysis and sensitivity analysis conducted in this study?

4. Results

a. Line 169-170: please provide explicit definitions for the pre-defined vegetarian diet and semi-vegetarian diet used in this study?

b. Line 179: kindly review the text and add the appropriate units where necessary

6. PLOS authors have the option to publish the peer review history of their article (what does this mean?). If published, this will include your full peer review and any attached files.

Reviewer #1: No

Reviewer #2: No

Reviewer #3: No

---

## [Author Response · Author response to Decision Letter 0]

25 Jul 2023

Dear Editor Kent Lai and Reviewer #1, Reviewer #2, and Reviewer #3,

Thank you so much for your work on our manuscript. Please accept our sincerely appreciation for your comments. On behalf of all authors in the article, I am sure that the scientific content of the article fully complies with all the journal requirements. Please read our answers about every question as following. We hope that our answers and corrections will be fine.

For Editor's comments:

Thank you for submitting your manuscript to PLOS ONE. After careful consideration, we feel that it has merit but does not fully meet PLOS ONE’s publication criteria as it currently stands. Therefore, we invite you to submit a revised version of the manuscript that addresses the points raised during the review process.

Our answer,

Thank you so much for your work on our manuscript. Please accept our sincere appreciation. We had carefully revised related content in our manuscript. The response of each question had been carefully answered in the part of “For Reviewer’s comments” as following. Please see the new manuscript and Figures.

1. Please ensure that your manuscript meets PLOS ONE’s style requirements, including those for file naming.

Our answer,

We had carefully checked our manuscript. And then, we corrected file naming of Figure so that controlling the its length within 15 words, and revised the footnote of Table 1 and cell lines of Table 2 for ensuring that our manuscript meets PLOS ONE’s style requirements. Please see the new manuscript and Figures.

Our answer,

We were very sorry that we did not do better in in providing match ‘Funding Information’ and ‘Financial Disclosure’. This research was funded by Joint Funds for the innovation of science and Technology, Fujian province (Grant number: 2020Y9133) and Fujian Maternity and Child Health Hospital (Grant number: YCXH 22-02). The funders had no role in study design, data collection and analysis, decision to publish, or preparation of the manuscript. We were sure that we had provided the correct grant numbers of funding, and had clarified financial disclosure.

3. Thank you for stating the following in your Competing Interests section: “The authors declare that they have no competing interests.” Please complete your Competing Interests on the online submission form to state any Competing Interests. If you have no competing interests, please state “The authors have declared that no competing interests exist.”, as detailed online in our guide for authors at http://journals.plos.org/plosone/s/submit-now. This information should be included in your cover letter; we will change the online submission form on your behalf.

Our answer,

We were sure that: “The authors declare that they have no competing interests”. 

Our answer, 

We had carefully reviewed and checked our reference by searching in Web of Science, Pubmed and Wangfang database one by one. We are sure that all references are complete and correct. And we did not cite any papers that have been retracted. 

5. Some of the figures were quite blurry. Please improve the resolution.

Our correction,

We were very sorry that we did not do better in processing resolution of Figures. We had improved the resolution of every Figure according to Figure format guidelines of PLOS ONE. Please see the new Figures.

For reviewer #1’s comments:

1. Line 50: Led to should be “leads”.

2. Line 52: Macrosomia and large for gestational age seem redundant; perhaps pick one to mention.

Our correction,

Page 3 line 46 to 49: GDM affects about 5.8~12.9% of pregnancies globally depending on different diagnostic criteria[2, 3] and leads several adverse obstetrical outcomes including hypertensive disorders, cesarean section, neonatal hypoglycemia, and macrosomia [4-6].

2. Zhu Y, Zhang C. Prevalence of Gestational Diabetes and Risk of Progression to Type 2 Diabetes: a Global Perspective. Curr Diab Rep. 2016;16(1):7.DOI: 10.1007/s11892-015-0699-x.

3. Chiefari E, Arcidiacono B, Foti D, Brunetti A. Gestational diabetes mellitus: an updated overview. Journal of endocrinological investigation. 2017;40(9):899-909.DOI: 10.1007/s40618-016-0607-5.

4. Ye W, Luo C, Huang J, Li C, Liu Z, Liu F. Gestational diabetes mellitus and adverse pregnancy outcomes: systematic review and meta-analysis. BMJ (Clinical research ed). 2022;377:e067946.DOI: 10.1136/bmj-2021-067946.

5. Davenport MH, Ruchat SM, Poitras VJ, Jaramillo Garcia A, Gray CE, Barrowman N, et al. Prenatal exercise for the prevention of gestational diabetes mellitus and hypertensive disorders of pregnancy: a systematic review and meta-analysis. Br J Sports Med. 2018;52(21):1367-75.DOI: 10.1136/bjsports-2018-099355.

6. Koivunen S, Torkki A, Bloigu A, Gissler M, Pouta A, Kajantie E, et al. Towards national comprehensive gestational diabetes screening - consequences for neonatal outcome and care. Acta Obstet Gynecol Scand. 2017;96(1):106-13.DOI: 10.1111/aogs.13030.

3. Line 292: adversely might be changed to inversely.

Our correction,

Page 20 line 325 to 327: In this systematic review and meta-analysis, we included ten studies with moderate to high quality involving fourteen records and found that greater adherence to plant-based dietary patterns is inversely associated with the risk of developing GDM.

For Reviewer #2’s comments:

It would have been nice if the composition of the plant based diets were examined in greater detail (for example, the breakdown of carbohydrate, protein and fat). Many plant based diets tend to be rich in carbohydrates which increase the risk of type 2 diabetes. In this context, if the authors could comment on why plant based diets are protective and how they differ from animal based diets, that would make the paper more interesting. In the “Discussion” on Page 22 and in the “Conclusion”, they do talk about this. But perhaps in the context of healthy carbohydrates and unhealthy carbohydrates, this section can be expanded a little bit more.

Our answer,

Thank you so much for your work on our manuscript. According to your comments, we carefully reviewed the details of plant-based dietary patterns proposed by the included studies. Regretfully, the included studies did not analyze plant-based dietary patterns in view of the breakdown of carbohydrate, protein and fat. Thus, we may not be able to examine this facet in greater detail. However, in “Discussion” section, we discussed the carbohydrates quality, which demonstrated the necessity of distinguishing between healthy carbohydrates and unhealthy carbohydrates. Additionally, we highlighted the importance of the source of protein and fat (from animal foods or plant foods) in plant-based dietary patterns. We hope that our discussion will be fine.

Our correction,

Page 22 line 380 to page 23 line 402: The stronger pooled RR was observed when we considered “healthful plant-based dietary indices” (fixed-effects, RR=0.86) instead of “unhealthy plant-based dietary indices” (fixed-effects, RR=0.90) in an included study[36]. The finding indicated an opinion that not all plant-based diets were beneficial, which has been emphasized in prior studies[61, 62]. Carbohydrates quality, a multidimensional index, usually be represented as some characteristics: glycaemic index, glycaemic load and dietary fiber content. It was considered as a crucial facet of plant-based diets[63]. For instance, plants with high glycaemic index, including potatoes, rice and refined bread, induced a sharp increase in glycaemia level that manifested the lower carbohydrates quality[64]. Plant-based diets dominated by such components were associated with higher risk of developing GDM[65, 66], whereas plant-based diets with low glycaemic index were related to the lower prevalence of GDM[67]. Besides, Looman et al.[62] unexpectedly found that low-carbohydrate diets, characterized by high fat, high protein and low carbohydrates, were positive associated with the risk of developing GDM. Their further exploration revealed that women with higher fiber and fruits intake had a decreased risk of developing GDM, however, women with higher cereal intake had increased risk of developing GDM. It also underscored the importance of the carbohydrates source. Except for carbohydrates quality, the source of protein and fat in plant-based dietary patterns mattered, too[68, 69]. Pregestational low-carbohydrate diets with high protein and fat from animal foods were positively associated with the risk of developing GDM, nevertheless, low-carbohydrate diets with high protein and fat from vegetable foods were inversely related to the risk of developing GDM[70]. In detail, higher animal protein intake would increase the risk of developing GDM, whereas higher plant protein intake would do not[68]. In addition, higher fat intake, whether animal fat or plant fat, would increase the risk of developing GDM[69]. As such, plant-based dietary patterns with higher carbohydrates quality and plant protein consumption were more encouraged. Moreover, the concept of healthful plant-based diet was proposed by Satija et al.[33], who classified sugar-sweetened beverages, sweets and desserts, refined grains and juices as relatively unhealthy plant diets. These food groups were related to a higher risk of GDM[71], which may be interpreted by following potential mechanisms. Higher glycemic load from sugar-sweetened beverages and sweets and desserts may lead to increased weight gain through decreasing satiety or increased hunger signals, and disrupt metabolism via affecting neurochemical changes in the brain[72-74]. Milling and squeezing during the productive process of refined grains and juices reduced dietary fiber and bioactive phytochemicals, then potentially lowering dietary quality[75]. Significantly, it is necessary to distinguish whether the type of juice is homemade or commercial. The homemade juices might not be removed the pulp and edible skin, thus can provide some essential components for preventing GDM, such as fiber, vitamins, minerals, and phytochemicals[76]. Adversely, commercial juices provide only excessive sugars and energy that overload the glycometabolism. Overall, only healthy plant-based diets reduce the risk of GDM rather than unhealthy plant-based diets. We recommended that quality of foods should be emphasized when sticking to plant-based dietary patterns.

33. Satija A, Bhupathiraju SN, Rimm EB, Spiegelman D, Chiuve SE, Borgi L, et al. Plant-Based Dietary Patterns and Incidence of Type 2 Diabetes in US Men and Women: Results from Three Prospective Cohort Studies. PLoS Med. 2016;13(6):e1002039.DOI: 10.1371/journal.pmed.1002039.

36. Chen ZL, Qian F, Liu G, Li MY, Voortman T, Tobias DK, et al. Prepregnancy plant-based diets and the risk of gestational diabetes mellitus: a prospective cohort study of 14,926 women. Am J Clin Nutr. 2021;114(6):1997-2005.DOI: 10.1093/ajcn/nqab275.

61. Satija A, Malik V, Rimm EB, Sacks F, Willett W, Hu FB. Changes in intake of plant-based diets and weight change: results from 3 prospective cohort studies. The American journal of clinical nutrition. 2019;110(3):574-82.DOI: 10.1093/ajcn/nqz049.

62. Looman M, Schoenaker D, Soedamah-Muthu SS, Geelen A, Feskens EJM, Mishra GD. Pre-pregnancy dietary carbohydrate quantity and quality, and risk of developing gestational diabetes: the Australian Longitudinal Study on Women's Health. The British journal of nutrition. 2018;120(4):435-44.DOI: 10.1017/s0007114518001277.

63. Filardi T, Panimolle F, Crescioli C, Lenzi A, Morano S. Gestational Diabetes Mellitus: The Impact of Carbohydrate Quality in Diet. Nutrients. 2019;11(7):1549.DOI: 10.3390/nu11071549.

64. Brouns F, Bjorck I, Frayn KN, Gibbs AL, Lang V, Slama G, et al. Glycaemic index methodology. Nutrition research reviews. 2005;18(1):145-71.DOI: 10.1079/nrr2005100.

65. Bao W, Tobias DK, Hu FB, Chavarro JE, Zhang C. Pre-pregnancy potato consumption and risk of gestational diabetes mellitus: Prospective cohort study. BMJ (Online). 2016;352:h6898.DOI: 10.1136/bmj.h6898.

66. Guo F, Zhang Q, Jiang H, He Y, Li M, Ran JJ, et al. Dietary potato intake and risks of type 2 diabetes and gestational diabetes mellitus. Clin Nutr. 2021;40(6):3754-64.DOI: 10.1016/j.clnu.2021.04.039.

67. Walsh JM, McGowan CA, Mahony R, Foley ME, McAuliffe FM. Low glycaemic index diet in pregnancy to prevent macrosomia (ROLO study): randomised control trial. BMJ (Clinical research ed). 2012;345:e5605.DOI: 10.1136/bmj.e5605.

68. Yong HY, Mohd Shariff Z, Mohd Yusof BN, Rejali Z, Tee YYS, Bindels J, et al. Higher Animal Protein Intake During the Second Trimester of Pregnancy Is Associated With Risk of GDM. Frontiers in nutrition. 2021;8:718792.DOI: 10.3389/fnut.2021.718792.

69. Feng Q, Yang M, Dong H, Sun H, Chen S, Chen C, et al. Dietary fat quantity and quality in early pregnancy and risk of gestational diabetes mellitus in Chinese women: a prospective cohort study. The British journal of nutrition. 2022:1-10.DOI: 10.1017/s0007114522002422.

70. Bao W, Bowers K, Tobias DK, Olsen SF, Chavarro J, Vaag A, et al. Prepregnancy low-carbohydrate dietary pattern and risk of gestational diabetes mellitus: a prospective cohort study. The American journal of clinical nutrition. 2014;99(6):1378-84.DOI: 10.3945/ajcn.113.082966.

71. Shin D, Lee KW, Song WO. Dietary Patterns during Pregnancy Are Associated with Risk of Gestational Diabetes Mellitus. Nutrients. 2015;7(11):9369-82.DOI: 10.3390/nu7115472.

72. Chen L, Hu FB, Yeung E, Willett W, Zhang C. Prospective study of pre-gravid sugar-sweetened beverage consumption and the risk of gestational diabetes mellitus. Diabetes Care. 2009;32(12):2236-41.DOI: 10.2337/dc09-0866.

73. Berthoud HR. The neurobiology of food intake in an obesogenic environment. The Proceedings of the Nutrition Society. 2012;71(4):478-87.DOI: 10.1017/s0029665112000602.

74. Levine AS, Kotz CM, Gosnell BA. Sugars: hedonic aspects, neuroregulation, and energy balance. The American journal of clinical nutrition. 2003;78(4):834s-42s.DOI: 10.1093/ajcn/78.4.834S.

75. Zhu Y, Olsen SF, Mendola P, Halldorsson TI, Yeung EH, Granström C, et al. Maternal dietary intakes of refined grains during pregnancy and growth through the first 7 y of life among children born to women with gestational diabetes. The American journal of clinical nutrition. 2017;106(1):96-104.DOI: 10.3945/ajcn.116.136291.

76. Yong HY, Mohd Shariff Z, Mohd Yusof BN, Rejali Z, Tee YYS, Bindels J, et al. Beverage Intake and the Risk of Gestational Diabetes Mellitus: The SECOST. Nutrients. 2021;13(7):2208.DOI: 10.3390/nu13072208.

For Reviewer #3’s comments:

1. To enhance clarity, several points in the abstract require further clarification:

a. Line 37-39: revisit the statement as the statement implies that both unhealthy and healthy plant-based diets are associated with a lower risk of GDM, with the healthy plant-based diet showing a slightly stronger association.

Our answer,

Thank you so much for your work on our manuscript. We were so sorry that we did not do better in clarifying this section. We had carefully revised related content in our manuscript. Please see the new Manuscript.

Our correction,

In the section of Abstract 

Page2 line 35 to 38: The slightly stronger association between plant-based diets and the risk of developing GDM was found when healthy plant-based dietary pattern index was included in pooled estimate (RR=0.86[0.79 to 0.94], I2=8.3%), compared with that unhealthy one was included (RR=0.90[0.82 to 0.98], I2=8.3%).

b. Line 40-41: This statement should be included in the methodology section.

Our correction,

In the section of Abstract

Page 2 line 31 to 32: Methods: This systematic review was conducted following the checklist of PRISMA. Six electronic databases including PubMed, Embase, Web of Science, China National Knowledge Infrastructure, Wangfang, and VIP database were searched from inception to November 20, 2022. A fixed or random effect model was used to synthesize results of included studies. Then, subgroup analysis, meta-regression and sensitivity analysis were performed to assure the reliability and stability of the results.

2. Introduction:

a. While the author highlights the benefits of a plant-based dietary pattern and its positive effects on maternal and infant outcomes, it would be valuable to consider mentioning the current prevalence or proportion of individuals adopting such a dietary pattern. Including this information would help support the rationale for conducting this study and shed light on the relevance and potential impact of investigating the association between plant-based diets and gestational diabetes mellitus.

Our correction,

In the section of Introduction:

Page 4 line 70 to 76: Plant-based dietary pattern is an umbrella term that refers to a habit that emphasized foods derived from plant sources more and from animal foods less, such as vegetarian diets[20, 21]. Globally, the prevalence of adopting vegetarian diets varied according to different countries culture, whereas it was commonly less than 10%[22]. It indicated that current proportion of pregnant women who adhered to remains vegetarian diets may still relatively small. The Academy of Nutrition and Dietetics advocated that vegetarian diets for pregnant women could be nutritionally adequate in pregnancy and result in positive maternal and infant outcomes[23], and this viewpoint had got partial support from several previous systematic reviews[24-27]. Practically, women’s dietary patterns changed little from before to during pregnancy[28]. It is confirmed that plant-based dietary pattern is beneficial to decrease the risk of developing type 2 diabetes mellitus[29]. However, the conclusion that the effects of plant-based dietary patterns on developing GDM remains unclear. Therefore, in this systematic review and meta-analysis, we aimed to investigate the effects of plant-based diets on the risk of developing GDM.

20. Gibbs J, Cappuccio FP. Plant-Based Dietary Patterns for Human and Planetary Health. Nutrients. 2022;14(8):1614.DOI: 10.3390/nu14081614.

21. David LA, Maurice CF, Carmody RN, Gootenberg DB, Button JE, Wolfe BE, et al. Diet rapidly and reproducibly alters the human gut microbiome. Nature. 2014;505(7484):559-63.DOI: 10.1038/nature12820.

22. Craig WJ, Mangels AR, Fresán U, Marsh K, Miles FL, Saunders AV, et al. The Safe and Effective Use of Plant-Based Diets with Guidelines for Health Professionals. Nutrients. 2021;13(11):4144.DOI: 10.3390/nu13114144.

23. Craig WJ, Mangels AR. Position of the American Dietetic Association: vegetarian diets. Journal of the American Dietetic Association. 2009;109(7):1266-82.DOI: 10.1016/j.jada.2009.05.027.

24. Raghavan R, Dreibelbis C, Kingshipp BL, Wong YP, Abrams B, Gernand AD, et al. Dietary patterns before and during pregnancy and birth outcomes: A systematic review. Am J Clin Nutr. 2019;109:729S-56S.DOI: 10.1093/ajcn/nqy353.

25. Sebastiani G, Barbero AH, Borrás-Novel C, Casanova MA, Aldecoa-Bilbao V, Andreu-Fernández V, et al. The effects of vegetarian and vegan diet during pregnancy on the health of mothers and offspring. Nutrients. 2019;11(3):557.DOI: 10.3390/nu11030557.

26. Tan C, Zhao Y, Wang S. Is a vegetarian diet safe to follow during pregnancy? A systematic review and meta-analysis of observational studies. Critical reviews in food science and nutrition. 2019;59(16):2586-96.DOI: 10.1080/10408398.2018.1461062.

27. Schiattarella A, Lombardo M, Morlando M, Rizzo G. The Impact of a Plant-Based Diet on Gestational Diabetes: A Review. Antioxidants (Basel, Switzerland). 2021;10(4):557.DOI: 10.3390/antiox10040557.

28. Crozier SR, Robinson SM, Godfrey KM, Cooper C, Inskip HM. Women's dietary patterns change little from before to during pregnancy. The Journal of nutrition. 2009;139(10):1956-63.DOI: 10.3945/jn.109.109579.

29. Qian F, Liu G, Hu FB, Bhupathiraju SN, Sun Q. Association Between Plant-Based Dietary Patterns and Risk of Type 2 Diabetes: A Systematic Review and Meta-analysis. JAMA internal medicine. 2019;179(10):1335-44.DOI: 10.1001/jamainternmed.2019.2195.

3. Materials and methods:

a. Line 92: The statement “Vegetarian dietary patterns or vegan dietary patterns were also reckoned with” may not conform to typical scientific writing standards.

Our answer,

We were sorry about we used atypical terms. We reviewed the scientific writing terms and corrected “vegetarian dietary patterns” and “vegan dietary patterns” to “vegetarian diet” and “vegan diet” according to MeSH terms, respectively. 

Our correction,

In the section of Materials and methods:

Page 5 line 104 to 105: According to related studies that assessed dietary patterns[31, 32], three methods mainly were used to define the plant-based dietary patterns: (a) prior designated diets that avoidance of animal-based foods, such as vegetarian diet and vegan diet; (b) a previous plant-based dietary indices scores; and (c) a posteriori factor analysis to extract dietary pattern.

31. Storz MA. What makes a plant-based diet? a review of current concepts and proposal for a standardized plant-based dietary intervention checklist. Eur J Clin Nutr. 2022;76(6):789-800.DOI: 10.1038/s41430-021-01023-z.

32. Gan ZH, Cheong HC, Tu YK, Kuo PH. Association between Plant-Based Dietary Patterns and Risk of Cardiovascular Disease: A Systematic Review and Meta-Analysis of Prospective Cohort Studies. Nutrients. 2021;13(11):3952.DOI: 10.3390/nu13113952.

b. Line 89-95: It would be beneficial to provide explicit definitions of both vegetarian and vegan dietary patterns to ensure clarity in understanding their distinctions. While both dietary patterns involve a focus on plant-based foods, vegetarian diets typically exclude meat, poultry, and seafood, while vegan diets abstain from all animal-derived products, including dairy, eggs, and honey. By clearly defining these terms, readers will have a better understanding of the specific dietary restrictions and considerations associated with each pattern, which is crucial for interpreting the study findings accurately.

Our answer, 

We had provided explicit definitions of both vegetarian and vegan diets. Moreover, to enhance clarity of article, we separated a single part to explain assessment of plant-based dietary patterns in our study.

Our correction,

In the section of Materials and methods,

Page 4 line 90 to 92: The inclusion criteria were defined as following: (a) Population involving women with singleton pregnancy and without any acute or chronic diseases that impact dietary intake, such as cancer or kidney disease. (b) The exposure of interest was adherence to the plant-based dietary patterns, which generally were defined as diets that consuming higher consumption of planted-based foods and lower consumption or avoidance of animal-based foods. (c) The comparator depends on actual exposure. (d) The outcome of interest was the incident of GDM……

Page 5 line 102 to page 6 line 130: 

Assessment of plant-based dietary patterns

According to related studies that assessed dietary patterns[31, 32], three methods mainly were used to define the plant-based dietary patterns: (a) prior designated diets that avoidance of animal-based foods, such as vegetarian diet and vegan diet; (b) a previous plant-based dietary indices scores; and (c) a posteriori factor analysis to extract dietary pattern.

Especially, vegetarian diets typically included plant-based foods including grains, legumes, nuts, seeds vegetables and fruits, and excluded all kinds of animal-based foods such as meat, meat products, poultry, seafood, mollusks and crustaceans[25]. Vegetarian diets usually contain two main directions: (1) lacto-ovo-vegetarian diets, which excluded meat but included dairy products, eggs and honey; (2) vegan diets, which excluded meat, dairy products, eggs and honey[25]. In this study, non-vegetarian diets or omnivorous diets were considered as the comparators of vegetarian diets/vegan diets.

When turns to previous plant-based dietary indices scores, Satija et al.[33] created a concept of plant-based dietary pattern indices to assess the adherence to plant-based dietary patterns including overall plant-based dietary indices (PDI), healthy plant-based dietary indices (hPDI), and unhealthy plant-based dietary indices (uPDI). Scoring was done according to quantile with the lowest food consumption receiving 1 point and quantile with highest food consumption receiving 5 points. For PDI, positive scores were assigned to plant food groups, and reverse scores were assigned to animal food groups. For hPDI, positive scores were assigned to healthy plant food groups (include whole grains, fruits, vegetables, nuts, legumes, vegetable oils and tea/coffee), while reverse scores were assigned to animal food groups and unhealthy plant food groups. For uPDI, positive scores were assigned to unhealthy plant food groups (included refined grains, potatoes and sweets/desserts), whereas reverse scores were assigned to animal food groups and healthy plant food groups. By above definition, the lowest PDI, hPDI and uPDI quantile, which indicated the poorest insistence on the plant-based dietary patterns, were used as a comparator of highest PDI, hPDI and uPDI quantile, respectively. Also, for the included studies that used factor analysis to extract plant-based dietary patterns, the lowest quantile was used as a comparator.

25. Sebastiani G, Barbero AH, Borrás-Novel C, Casanova MA, Aldecoa-Bilbao V, Andreu-Fernández V, et al. The effects of vegetarian and vegan diet during pregnancy on the health of mothers and offspring. Nutrients. 2019;11(3):557.DOI: 10.3390/nu11030557.

31. Storz MA. What makes a plant-based diet? a review of current concepts and proposal for a standardized plant-based dietary intervention checklist. Eur J Clin Nutr. 2022;76(6):789-800.DOI: 10.1038/s41430-021-01023-z.

32. Gan ZH, Cheong HC, Tu YK, Kuo PH. Association between Plant-Based Dietary Patterns and Risk of Cardiovascular Disease: A Systematic Review and Meta-Analysis of Prospective Cohort Studies. Nutrients. 2021;13(11):3952.DOI: 10.3390/nu13113952.

33. Satija A, Bhupathiraju SN, Rimm EB, Spiegelman D, Chiuve SE, Borgi L, et al. Plant-Based Dietary Patterns and Incidence of Type 2 Diabetes in US Men and Women: Results from Three Prospective Cohort Studies. PLoS Med. 2016;13(6):e1002039.DOI: 10.1371/journal.pmed.1002039.

c. Line 109: Please verify the accuracy of the term ‘VIP’ as a database name. Please double-check its validity or provide additional clarification if it refers to a specific database or acronym.

Our answer,

‘VIP’ is the abbreviation of Chinese Scientific Journals Database. We have provided the complete description here for accuracy.

Our correction,

In the section of Abstract: 

Page 2 line 27 to 30: Methods: This systematic review was conducted following the checklist of PRISMA. Six electronic databases including PubMed, Embase, Web of Science, China National Knowledge Infrastructure, Wangfang, and Chinese Scientific Journals Database were searched from inception to November 20, 2022. A fixed or random effect model was used to synthesize results of included studies…….

In the section of Materials and methods:

Page 6 line 133 to 135: Six electronic databases including PubMed, Embase, Web of Science, China National Knowledge Infrastructure (CNKI), Wangfang, and Chinese Scientific Journals Database (VIP) were searched using a strategy of Mesh-term combined with text-word.

d. Please consider rechecking the PRISMA checklist to ensure its adherence to the guidelines. Upon review, it appears that certain items may not fully align with the recommended criteria. For example: Data items, list and define all outcomes for which data were sought. Specify whether all results that were compatible with each outcome domain in each study were sought (e.g. for all measures, time points, analyses).

Our correction,

We carefully rechecked items of the PRISMA checklist and refilled it. Please see our new S3 Appendix. 

e. Line 148-152: please provide additional elaboration on the details of the subgroup analysis and sensitivity analysis conducted in this study?

Our correction,

Page8 ling 175 to 187: Subgroup analysis and meta-regression analysis were estimated for regulating effect sizes of the following possible variables: study country (the United States, China, India, Australia and Malaysia), method of dietary patterns assessment (prior designated diets, calculated plant-based dietary indices scores and factor analysis to extract dietary pattern) and period of dietary patterns investigation (before pregnancy and during pregnancy). Sensitivity analysis was conducted by removing one study from overall analysis each time. Moreover, for studies that reported the results before and after BMI adjustment, we included those data to assess the pooled results changes before and after BMI adjustment. In the studies that defined adherence to plant-based dietary patterns using dietary overall or healthy/unhealthy plant-based dietary indices, the relationship for overall plant-based dietary indices was estimated in the pooled result evaluation, the relationship for healthful/unhealthy plant-based dietary indices were estimated in a sensitivity analysis. Besides, sensitivity analysis was conducted through excluding any single study.

4. Results

a. Line 169-170: please provide explicit definitions for the pre-defined vegetarian diet and semi-vegetarian diet used in this study?

Our answer, 

In this study, we used the definitions for the pre-defined vegetarian diet and semi-vegetarian diet were defined by the authors of the included study. We had added the definitions in the Manuscript. 

Our correction,

Page 9 line 203 to 207: Particularly, Yisahak et al.[35] reported two records that full vegetarian diet and semi-vegetarian diet from two investigations. They defined ‘full vegetarian diet’ as never ate meat, poultry and fish, or ate these foods less than once a month but had no restriction on fish; and ‘semi-vegetarian diet’ as ate meat, poultry and fish greater than once a month, but less than once a week.

35. Yisahak SF, Hinkle SN, Mumford SL, Li MY, Andriessen VC, Grantz KL, et al. Vegetarian diets during pregnancy, and maternal and neonatal outcomes. International journal of epidemiology. 2021;50(1):165-78.DOI: 10.1093/ije/dyaa200.

b. Line 179: kindly review the text and add the appropriate units where necessary

Our correction,

Page 9 line 215 to page 10 line 217: Totally, 32,006 women were included, who were with mean pre-pregnancy/pregnancy BMI ranging from 20.4 to 25.8 kg/m2 and their age ranging from 26.7 to 32.3 years old.

All the best.

Thanks again.

Yours sincerely,

Authors

---

## [Decision Letter · Decision Letter 1]

4 Sep 2023

The effects of plant-based dietary patterns on the risk of developing gestational diabetes mellitus: A systematic review and meta-analysis

PONE-D-23-15568R1

Dear Dr. Jiang,

We’re pleased to inform you that your manuscript has been judged scientifically suitable for publication and will be formally accepted for publication once it meets all outstanding technical requirements.

Kind regards,

Kent Lai

Academic Editor

PLOS ONE

Additional Editor Comments (optional):

Reviewers' comments:

Reviewer's Responses to Questions

**Comments to the Author**

1. If the authors have adequately addressed your comments raised in a previous round of review and you feel that this manuscript is now acceptable for publication, you may indicate that here to bypass the “Comments to the Author” section, enter your conflict of interest statement in the “Confidential to Editor” section, and submit your "Accept" recommendation.

Reviewer #2: All comments have been addressed

Reviewer #3: All comments have been addressed

2. Is the manuscript technically sound, and do the data support the conclusions?

Reviewer #2: Yes

Reviewer #3: Yes

3. Has the statistical analysis been performed appropriately and rigorously? 

Reviewer #2: Yes

Reviewer #3: Yes

4. Have the authors made all data underlying the findings in their manuscript fully available?

Reviewer #2: Yes

Reviewer #3: Yes

5. Is the manuscript presented in an intelligible fashion and written in standard English?

Reviewer #2: Yes

Reviewer #3: Yes

6. Review Comments to the Author

Reviewer #2: Dear Dr. Kent Lai,

Thank you for asking me to re-review the article entitled “PONE-D-23-15568R1 - The effects of plant-based dietary patterns on the risk of developing gestational diabetes mellitus: A systematic review and meta-analysis”. The authors have carried out all the suggestions given by me and the paper may now be Accepted for publication.

Thanking you and with regards,

Dr. V. Mohan

Reviewer #3: Thank you for addressing the comments. The authors have addressed all of my comments. I have no further comments.

7. PLOS authors have the option to publish the peer review history of their article (what does this mean?). If published, this will include your full peer review and any attached files.

Reviewer #2: No

Reviewer #3: No

---

## [Editor Report · Acceptance letter]

11 Sep 2023

PONE-D-23-15568R1 

The effects of plant-based dietary patterns on the risk of developing gestational diabetes mellitus: A systematic review and meta-analysis 

Dear Dr. jiang:

I'm pleased to inform you that your manuscript has been deemed suitable for publication in PLOS ONE. Congratulations! Your manuscript is now with our production department. 

Kind regards, 

on behalf of

Dr. Kent Lai 

Academic Editor

PLOS ONE